# Mechanical Performance of Steel Fibre Reinforced Concrete Exposed to Wet–Dry Cycles of Chlorides and Carbon Dioxide

**DOI:** 10.3390/ma14102642

**Published:** 2021-05-18

**Authors:** Victor Marcos-Meson, Gregor Fischer, Anders Solgaard, Carola Edvardsen, Alexander Michel

**Affiliations:** 1Department of Civil Engineering, Technical University of Denmark, 2800 Kongens Lyngby, Denmark; gf@byg.dtu.dk (G.F.); almic@byg.dtu.dk (A.M.); 2Department of Tunnels and Underground Infrastructure, COWI A/S, 2800 Kongens Lyngby, Denmark; adso@cowi.com (A.S.); cle@cowi.com (C.E.); 3VIA Building, Energy, Water & Climate, VIA University College, 8700 Horsens, Denmark

**Keywords:** steel fibre reinforced concrete (SFRC), corrosion, chlorides, carbonation, cracks, wet–dry cycles, exposure time

## Abstract

This paper presents an experimental study investigating the corrosion damage of carbon-steel fibre reinforced concrete (SFRC) exposed to wet–dry cycles of chlorides and carbon dioxide for two years, and its effects on the mechanical performance of the composite over time. The results presented showed a moderate corrosion damage at fibres crossing cracks, within an approximate depth of up to 40 mm inside the crack after two-years of exposure, for the most aggressive exposure conditions investigated. Corrosion damage did not entail a significant detriment to the mechanical performance of the cracked SFRC over the time-scales investigated. Corrosion damage to steel fibres embedded in uncracked concrete was negligible, and only caused formation of rust marks at the concrete surface. Overall, the impact of fibre damage to the toughness variation of the cracked composite over the time-scale investigated was secondary compared to the toughness variation due to the fibre distribution. The impact of fibre corrosion to the performance of the cracked composite was subject to a size-effect and may only be significant for small cross-sections.

## 1. Introduction

Steel fibre reinforced concrete (SFRC) is increasingly used in civil engineering as partial or total replacement of conventional reinforcing steel. Carbon-steel fibres (henceforth “steel fibres”) are being used, among others, for the construction of infrastructure exposed to corrosive environments [1,2,3]. However, the total replacement of conventional reinforcement with steel fibres is still controversial when considering the durability of cracked SFRC under corrosive exposures [4].

Former studies investigating the performance of SFRC exposed to chlorides and carbon dioxide reported limited corrosion damage for uncracked SFRC [4], which occurred mainly at the steel fibres adjacent to the concrete surface and led to negligible long-term detriment to the mechanical performance of the composite over exposure periods in the field up to 20 years to chloride [5,6,7,8,9] or carbonation exposure [7,10,11,12]. However, there is an open discussion regarding the corrosion of steel fibres bridging cracks in the range of 0.1–0.3 mm in SFRC under these exposures, and its impact on the residual mechanical performance of the cracked SFRC [4].

Field exposure of cracked SFRC to saltwater (i.e., exposure classes corresponding XS2-XS3 according to EN-206 classification) generally showed an early stabilization of the deterioration process during the first 2–3 years of exposure. The exposure entailed moderate corrosion damage during periods of 1–2 years [6,10,13], and even increased residual tensile strength over time for small crack widths [6,10,13,14,15,16]. Similarly, field exposure of cracked SFRC to rainwater (i.e., XC4 exposure class) presents contradictory results: showing minor corrosion damage and limited strength loss [10], or else, critical corrosion damage [13] for short-term exposures (i.e., 1–2 years). Longer exposure times (i.e., up to 5 years) to similar conditions resulted in limited corrosion damage for cracks in the abovementioned range [16]. Unfortunately, long-term exposure data (e.g., up to 20 years) is scarce and mostly comprises qualitative results from visual inspection for both chloride [7,16] and carbonation exposure [11,17].

Alternatively, the use of laboratory exposure to wet–dry cycles has been proven effective to accelerate the deterioration of cracked SFRC exposed to chlorides and carbon dioxide compared to field-exposure [4], by a factor of around 1:7 [11] to 1:50 [16] for the same exposure times. However, experimental studies under laboratory conditions often show contradictory results, which are reported to be very sensitive to variations in various parameters: the specimen dimensions [16,18], exposure conditions [8,16,19], and exposure time [4].

Accelerated exposure to chlorides under wet–dry cycles for short times (e.g., up to 6 months) generally resulted in negligible toughness loss for cracked SFRC [16,20]; however, some studies show contradicting results [6,21,22]. While extended exposures (e.g., 1–3 years) to corresponding conditions also showed discrepant results: entailing minor corrosion damage [19,23] or else a large deterioration [24,25]. Similarly, laboratory exposure of cracked SFRC (0.2–0.5 mm cracks) to wet–dry cycles of freshwater and high CO_2_ concentrations over 18 months resulted in minor corrosion damage of the cracked composite [20], as also observed in pre-carbonated specimens under similar conditions [11].

### Scope and Research Significance

This paper investigates the corrosion damage on cracked SFRC exposed to wet–dry cycles of chlorides and carbon dioxide, and its impact on the mechanical performance of the uncracked and cracked SFRC. The investigation covers the exposure, mechanical testing and inspection of bending and uniaxial tension specimens cracked at 0.15 and 0.3 mm and exposed to wet–dry cycles after one- and two-year exposure. The discussion presented in this paper focuses on describing the extent of fibre corrosion over the exposure time and its subsequent impact on the mechanical performance of the cracked and uncracked SFRC, based on a statistical analysis of the test data.

The investigations herein were performed with focus on typical engineering applications of structural SFRC, e.g., prefabricated segmental linings; therefore, the mix-design and fibre type and content used were chosen as representative of (e.g., a concrete with a characteristic compressive strength class of C50 and a characteristic residual strength class of 4C at 28-days).

The discussion herein aims at providing a detailed background dataset describing the mechanical performance of SFRC (with focus on cracked SFRC) under corrosive exposures (e.g., chlorides and carbon dioxide). This dataset, together with former studies, may serve as a basis for the development of future design codes for structural SFRC, that may also cover the design of cracked SFRC under aggressive corrosive exposures.

## 2. Methodology

This investigation covers the preparation, exposure, mechanical testing, and inspection of ca. 420 SFRC specimens, with induced crack widths of 0.15 and 0.3 mm and exposed to wet–dry cycles during years. Specimens were tested after one- and two-years of exposure. The exposures investigated comprise chloride and carbon dioxide exposure in wet–dry cycles. Two types of specimens were investigated: three-point bending notched beams according to EN 14651:2007 [26] and single notched coupon tests in uniaxial tension based on the experiments presented in [27].

The analysis of the mechanical performance of the material is made by the comparison of the stress vs Crack Mouth Opening Displacement (i.e., stress–CMOD) and work–CMOD response (i.e., the energy absorption) for the investigated material in bending and uniaxial tension. Visual inspection of the crack surface and fibres bridging the crack were used to determine the degree of fibre damage due to corrosion. Experimental results are shown in Appendix A and Appendix A.

Finally, discussions regarding the extent of fibre corrosion over time and its impact on the residual performance of the cracked SFRC are based on descriptive statistics of the fibre distribution and toughness data of the cracked composite. Furthermore, the relative impact of the main experimental variables on the mechanical performance of the cracked composite over time is discussed based on a regression analysis.

The results and discussion section herein use statistical terminology, pretraining the use of the words “specimen” and “sample”:
(i)“Specimen” refers to a single sampling unit (i.e., each one of the tests executed). Separate specimens are not compared, unless explicitly specified in the text.(ii)“Sample” refers to a group of specimens exposed to the same environment and cracked at the same CMOD (i.e., a sample is a group of 9 specimens for the bending tests and 10 specimens for the uniaxial tension tests). The discussion is based on comparison of samples.

### 2.1. Preparation of Specimens

The specimens were prepared following a mix-design in compliance with the recommendations for minimum binder content and water-to-binder ratio specified for conventional reinforced concrete in DS/EN 206-1:2011 for the exposure classes XC4–XS3. The total binder content was 426.3 kg/m^3^ with 31% fly ash replacement of the Portland cement. The water to binder ratio was 0.34 and the equivalent water to cement ratio was 0.40, considering an effective k-factor for the fly ash of 0.40, see Table 1. The superplasticizer and air-entrainer content were adjusted in the subsequent mixes to reach a slump of 100 ± 20 mm and an entrained air content of 4.5 ± 0.5%, measured according to EN 12350-2 and EN 12350-7, respectively.

The steel fibre used was a cold-drawn hooked-ended fibre (type 1 according to EN 14889-1:2006 [29]), with a length of 60 mm and diameter of 0.75 mm (aspect ratio: l/d = 80). The fibre was made of high-carbon cold-drawn steel with a characteristic ultimate tensile strength of 1900 MPa (as per producer’s specification).

The production of the bending specimens was done in a prefabrication plant, using an industrial mixing plant. The casting was made by direct pumping of the concrete on coated plywood forms over an industrial vibration table. The beam elements (600 × 150 × 150 mm) were filled from one end in two steps and were vibrated for 4 min in total. The specimens were cast in 3 separate batches in consecutive days, demoulded after one day and cured indoors, moist covered with plastic foil for 56 days at 20 °C.

The uniaxial tension specimens were cast in the laboratory with a 300 L planetary mixer. The specimens were cast in 600 × 150 × 150 mm steel formworks with built-in lateral grooves (see Figure 1b), over a vibration table in two steps and were vibrated in total for 4 min at 50 Hz. The specimens were cast in 5 separate batches in consecutive days, demoulded after one day and cured indoors, moist covered with plastic for 56 days at 20 °C. The specimens were cut in cubes after 28 days of curing and thereafter cured for additional 28 days.

The final dimensions of the three-point bending beam specimens were 600 × 150 × 150 mm, with a 25 mm deep and 5 mm thick notch cut at the centre, along the transversal direction, according to [26], see Figure 1a. The effective cross-section at the notch was 150 × 125 mm. The dimensions of the uniaxial tension cube specimens were 150 × 150 × 150 mm, with a 35 mm deep and 5 mm thick notch cut along the perimeter, leaving an effective cross-section of 80 × 80 mm inside the notch, as shown in Figure 1b.

The bending specimens were grouped in samples of 9 replicates and the uniaxial tension specimens were grouped in samples of 10 replicates. The specimens were distributed uniformly in the samples from the batches they were casted. Cubes for additional compression tests were cut off the last 150 mm of the bending beams after testing and 10 replicates were tested for every exposure.

### 2.2. Exposure Setup

Table 2 describes the exposure environments, that comprise wet–dry cycles of four days (i.e., of two days each). The test-groups (i.e., samples) were coded as follows: (w) crack width; being 0.15 and 0.3 mm; (s) salinity of wet cycle, being 3.5% and 7.0%; and (c) carbon dioxide concentration, being 0.05%vol. for ambient exposure and 0.5%vol for accelerated carbonation exposure. The exposure time for the specimen was marked with a letter: (A) for one-year exposure and (B) for two-year exposure.

Two reference scenarios were tested: (i) uncracked samples that were kept covered (w0s0c0t0 and w0s0c0B), and (ii) cracked samples that were exposed to wet–dry cycles of limewater and air (w15s0c0 and w30s0c0). The cracked samples exposed to corrosive environments were divided into four exposures, shown in Table 2.

The exposure setup consisted of 10 polyethylene containers of 1 m^3^, connected in pairs and providing five different exposures, see Figure 2. Each pair of containers was connected with two membrane pumps that circulated the solution to the tank running the wet cycle, while specimens in the other tank were exposed to the dry cycle in the meantime. The cycles were operated automatically from an electronic controller. The exposure solution (ca. 500 L) was pumped at a rate of 4.5 L/min, covering the specimens by ca. 20 cm of solution.

For the air exposed specimens, the drying cycle was provided by a fan (100 mm diameter) with a nominal flow of 93 m^3^/h, installed at the centre of the upper side of the tank. The fan created an air flow towards four outlets of 100 mm diameter placed at the top four corners (see Figure 2) and was mixed with the laboratory air. The ventilation system of the laboratory kept the air at a stable temperature and humidity. The drying cycle of the carbon dioxide exposure ran through a closed loop and utilized a cooled heat exchanger to condensate moisture from the air flux before the inlet (see Figure 2); the nominal air flow was 93 m^3^/h.

The specimens were placed in vertical position, i.e., with the crack horizontally, with a separation of ca. 50 mm between specimens that ensured the circulation of air inside the tank. The beam specimens were placed with an ca. 5° inclination, leaving the crack mouth facing upwards to facilitate the release of entrapped air inside the crack.

The exposure solution was substituted with the following programme: (i) every two weeks during the first three months of the exposure, (ii) every month until 6 months of exposure, and (iii) every two months after six-months and until the end of the exposure (2-years). The solution of the cracked reference samples (w15s0c0 and w30s0c0) was not replaced, instead a 20/80 mix of saturated NaOH and CaOH was added every week. The pH value in the solution was kept in the range of 10–13.5. The exposure media for the specimens exposed to the carbon dioxide cycles (w15s0c5 and w30s0c5) was non-chlorinated fresh water (pH = 7.5–8.0, Cl^−^ < 50 mg/L, 13–15 °dH). The same type of water was also used to prepare the saltwater solutions.

The composition of all the exposure solutions was checked by Total Dissolved Solids (TDS) and pH measurements weekly. The concentration of Cl^-^ in the saltwater solutions was measured before replacing the solution by spectrophotometry [30] and was compared against the TDS values.

The relative humidity and temperature inside the laboratory were monitored, being ca.: 20 ± 2 °C and 50 ± 10%, respectively. The CO_2_ concentration of the laboratory air and inside the CO_2_ closed circuit were measured weekly and were 500 ± 100 ppm and 5000 ± 1000 ppm respectively.

### 2.3. Mechanical Testing

The experiments were performed in the following order: (i) testing of the reference sample of specimens at 56 days; (ii) cracking of all the remaining specimens at 56 days and preparation for exposure; (iii) exposure for one- and two-years; (iv) testing after one year of exposure; and (v) testing after two years of exposure.

The bending tests were done in a 100 kN flexural test frame, according to EN 14651 [26]. The Crack Mouth Opening Displacement at the end of the notch (CMOD_N_) was measured at the centre of the notch with a clip-gage connected to two steel pins glued to the face of the notch (Figure 1a).

The uniaxial tension tests were performed in a 500 kN test-frame. The uniaxial-tension setup consisted of two steel grips hydraulically clamped at the two ends of the test frame, based on the setup described in [27]. The test rig transferred the tensile load through the two indentations placed at the sides of the specimen; while the two steel grips were coupled with four sliding steel rails at the corners of the rig to restrain rotation and torsion at the specimen during the test. The CMOD_N_ was measured by two clip gages, each with a total travel length of 5 mm, connected to two steel pins glued to the centre of opposite faces (see Figure 1b).

The compression tests were executed in a 4000 kN capacity compression frame, according to the specifications of EN 12390-3:2012 [31].

The testing of the residual-flexural and residual-tensile strength of the specimens was done in accordance with the displacement rates specified in [26], with a sampling frequency rate of 100 Hz. After reaching a crack width of 5 mm, the displacement rate was increased up to 1 mm/min, until the specimen was split open completely.

The specimens were cracked before exposure with cracks of 0.15 and 0.3 mm calculated at the crack mouth (CMOD_M_). After the target crack width was reached, the displacement of the crosshead was locked, and the crack was supported with High-Density Polyethylene (HDPE) inserts inside the notch. After the preparation tests, the opening at the crack mouth for the bending specimens was CMOD_M_ = 0.10 ± 0.01 and 0.25 ± 0.03 mm for the specimens cracked at 0.15 mm and 0.3 mm, respectively; and CMOD_N_ = 0.13 ± 0.03 and 0.25 ± 0.04 mm for the uniaxial tension specimens cracked at 0.15 mm and 0.3 mm, respectively. For simplicity, crack widths will refer only to the target crack opening at the crack mouth hereafter, i.e., 0.15 and 0.3 mm CMOD_M_.

#### Processing of Data from Experiments

First, the test data collected from each specimen (i.e., the load-CMOD_N_) was resampled to a resolution of 1 µm CMOD_N_. Second, the resamples load-CMOD_N_ data was filtered and smoothed, using a median filter (block size of 5) and a moving average filter (block size of 3).

The opening displacement at the crack mouth (CMOD) for the bending specimens was calculated from the CMOD_N_ measurements based on inverse calculation of the neutral axis position applying the cracked-hinge model [32]. The CMOD values for the uniaxial tension specimens was calculated as the mean value from the two CMOD_N_ measurements registered. Hereafter, crack opening values discussed in this paper will only refer to the opening displacement at the crack mouth (CMOD).

The residual-flexural and -tensile strengths were calculated for the bending and uniaxial tension tests, respectively. The calculation of the residual flexural strength was done according to EN 14651 [26], considering that: there is a single crack, that initiates at the notch and propagates perpendicular to the length of the beam, which covers the entire cross-section. The residual tensile strength of the uniaxial tension tests was calculated as the ratio of the load and cross-section, assuming a single crack with a uniform crack width.

The energy absorbed by the system during the test, denoted as “work”, was calculated as the integration of the area below the load–CMOD curve for both bending and uniaxial tension specimens.

Finally, a lognormal probability distribution was fitted to each sample of data (i.e., group of specimens), the experimental results are presented herein (see Appendix A) as the mean value of the load at each CMOD value, together with the upper- and lower- confidence bounds at 90% confidence interval (90% CI).

### 2.4. Visual Inspection and Fibre Counting

After finalizing the mechanical tests, the crack was completely opened. Each of the fibres crossing the crack were classified according to the degree of corrosion damage observed at the fibre and counted, following an approach similar to [11] (after the method shown in), see Figure 3. The fibres marked according to the following categories:
Level-1, no corrosion (green): fibres with no corrosion damage visible at ×2 magnification. Under the assumption that the fibre contributed to the residual mechanical performance of the specimen, see Figure 3a.Level-2, minor corrosion (yellow): fibres with rust stains at the surface that do not show a visible cross-section loss at ×2 magnification. Presuming that there was no impact of corrosion on the fibre influence to the residual tensile performance of the specimen, see Figure 3b.Level-3, moderate corrosion (orange): presence of pits and visible reduction of cross-section at ×2 magnification, ca. 10–30%. Presuming that fibre corrosion had an impact on the fibre contribution to the residual tensile performance of the specimen, see Figure 3c.Level-4, severe corrosion (red): presence of large pits and total or major loss of cross-section, e.g., cross-section loss larger than ca. 30%. Presuming that the fibre had no contribution to the residual mechanical performance of the specimen, see Figure 3d.Fibre rupture, (blue): additional indicator to mark fibres that ruptured instead of pulled-out, see Figure 3a,c. Fibre rupture was always assumed for fibres with major corrosion (level 4), see Figure 3d.

The fibre counting was made by visual inspection, placing acrylic modelling paste of selected colours at the intersection of each fibre with the crack face, as shown Figure 4a. Afterwards, a high-resolution image of the surface of the open crack was taken and analysed in batches using an image analysis algorithm: the location and classification of each fibre was calculated by means of colour segmentation on the HSV representation of the image and subsequent calculation of the centroids of each point in the mask, see an example in Figure 4b.

Colorimetric tests were done on the cracked surface of the specimens to estimate: (i) the depth of the penetration front of chlorides at the crack faces spraying 0.1N AgNO_3_; and the carbonation depth inside the crack, spraying a 1 wt.% phenolphthalein solution (pH threshold ≈9) and rainbow indicator (pH thresholds in the range: 5–7–9–11–13). These tests were used to confirm whether there was ingress of chlorides and carbonation damage inside the crack but were not intended to quantify the chloride concentration or pH inside the crack. An assessment of these results was published in [28].

The processed data is presented in Appendix B as discrete contours of the total density of fibres for each sample, i.e., showing the average number of fibres per dm^2^ of each group of specimens.

### 2.5. Statistical Analyses

The results section includes two main types of statistical analysis: comparison of samples based on the Student’s *t*-test, used in Section 3.1, Section 3.2 and Section 3.4, and regression modelling used in Section 3.5.

#### 2.5.1. Comparison of Samples

The comparison of samples, shown in Section 3.1, Section 3.2 and Section 3.4, was performed by the Student’s *t*-test (using the Welch’s approximation for samples with unequal variance) [33]. The test calculates the probability (*p*-value) for the null hypothesis (H_0_) being true. The null hypothesis (H_0_) corresponds to the assumption that the two distributions compared have the same mean value. Probabilities (*p*-values) for the null hypothesis (H_0_) which are lower than the level of significance (α), indicate that the null hypothesis (H_0_) may be considered “not-significant”; so that the alternative hypothesis (H_a_) may be correct. If so, the samples have a statistically-significant higher or lower mean value than the reference within a (1 − α) confidence.

In Section 3.1, a two-tailed *t*-test is used to test the alternative hypothesis (H_a_) that the mean value of the compressive strength for the exposed sample being different than the unexposed reference samples. The level of significance was set to α = 10%.

In Section 3.2, the mean values of the work–CMOD curves for the bending and uniaxial tension samples are compared among themselves and to the references by a one-tailed *t*-test (i.e., for both the right and left tails). Two alternative hypotheses (H_a_) were tested: (i) the probability of the mean value of the sample tested after two years being greater (right tail) or smaller (left tail) than the unexposed reference sample tested after 56 days; (ii) the probability of the mean value of the sample tested after two years of exposure being greater (right tail) or smaller (left tail) than its corresponding sample tested after one year of exposure. The level of significance was set to three levels: i.e., α = 5, 15, and 25%.

In Section 3.4, the mean value of the ratio of ruptured fibres is compared by a one-tailed Welch’s *t*-test: (i) to the mean value of the uncorroded reference (Level 1) for each exposure time; and (ii) to the mean values of each group, calculated for the samples tested after one-year exposure. The level of significance was set to α = 10%.

#### 2.5.2. Regression Model

The effect of the main study variables on the mechanical performance of the cracked SFRC were evaluated based on Multiple Linear Regression (MLR) analysis in Section 3.5.

The MLR analysis was used to quantify the contribution of the main parameters to the variation in the toughness of the cracked SFRC tested after one- and two-years exposure. The regression model covers the main effects and 2-factor interactions of the following variables: fibre content (x1), ratio of corroding fibres for levels 2–4 (x2–4), ratio of ruptured fibres (x5), the crack width (x6), and the exposure time (x7). The independent and response variables were standardized.

Specifically, the model described the correlation of the fibre content, fibre damage (i.e., corrosion and rupture), crack width, exposure conditions and exposure time to the variation in toughness of the material (y), expressed as the total work at a crack opening 0.5–4.0 mm in bending and tension. The initial model is described in Equation (1) in Wilkinson notation [34]:y ~ i + x1:x2:×3:x4:x5:x6:x7(1)

The predictor coefficients were fitted to the data using robust regression, i.e., least trimmed squares (LTS) with a bisquare weight function for the residuals. The model was reduced iteratively by backwards component selection applied to the interaction terms (threshold α = 10%); while the main terms of the models were not reduced.

## 3. Results

The results presented below focus on describing the role of the exposure time in the main aspects that describe the deterioration of SFRC: (i) the strength of exposed uncracked SFRC (Section 3.1); (ii) the toughness of exposed cracked SFRC (Section 3.2); (iii) the extent and severity of fibre corrosion (Section 3.3); (iv) the ratio of fibres rupturing and its relation to fibre corrosion (Section 3.4); and (v) the relative impact of the main variables affecting the toughness of the cracked material (Section 3.5).

The results shown below are based on the analysis of the processed experimental data presented in: Appendix A for the mechanical results, and Appendix B for the fibre count results. Furthermore, the experimental data is published in a tabulated format in [35].

### 3.1. Variation of the Strength of Uncracked SFRC

The compression test results are presented in Figure 5 for the tests performed after one- and two-years of exposure. The unexposed reference samples (w0s0c0t0 and w0s0c0A/B) are presented together in Figure 5a as “REF”. The data is also presented classified by the age of the specimen in months, see Figure 5b; and by the concrete batch from which the specimens were produced (batch A–C), see Figure 5c.

The mean values of the samples grouped by exposure are similar to each other, within 72–75 MPa, as shown in Figure 5a, and indicate negligible change of the compressive strength comparing the exposures. However, the results showed a significant scatter within each sample, with an average standard deviation close to 8 MPa (i.e., ca. 10% of the mean value), related to production variations between the three batches (batch A–C), as shown in Figure 5c. The results presented in Figure 5a were also compared to the unexposed reference samples (i.e., shown combined as “REF”) by a two-tailed Welch’s *t*-test (e.g., statistical significance is considered for α = 10%). The results of the *t*-test showed a statistically non-significant variation of the mean values.

### 3.2. Variation of the Residual Mechanical Performance of Cracked SFRC

This section presents a study of the variations of the tensile toughness of the cracked composite due to the exposure over time (i.e., after two years exposure), based on the data presented in Appendix A. In this section, the data collected after two-year exposure is compared against reference samples (i.e., uncracked and cracked references) and the data after one-year exposure.

The toughness of the cracked material was described by the total energy released during the tests (i.e., work); and was calculated below as the integral of the load-CMOD curve in the range of CMOD = 0.5–4.0 mm. The work values at 4 mm CMOD after one- and two-years exposure are presented as a boxplot in Figure 6.

The data presented in Figure 6 shows an overview of the experimental results, displaying a generally large scatter of the toughness values, regardless of the crack width, exposure or age. The indication of statistically significant variations of the material toughness is discussed in the paragraphs below.

The significance of the changes in the material toughness (i.e., work) as a function of CMOD values was evaluated comparing the mean values of the work-CMOD curves for the bending and uniaxial tension samples to various references by means of a one-tailed *t*-test, as shown in Figure 7. The figures show the probability (*p*-value) of: (a) the mean value of the sample being smaller than the reference (left tail); (b) the mean value of the sample being greater than the reference (right tail). The significance thresholds (Alpha) considered for the alternative hypotheses (Ha) are: 95, 85, and 75%, that are displayed as dotted horizontal lines.

In short, *p*-values at the top side of the graph indicate that the corresponding samples had a higher mean toughness compared to the reference values, while *p*-values at the bottom side indicate lower toughness; being more probable if closer to the top (or bottom) edges.

The distribution of the work-CMOD curves of the bending and uniaxial tension samples exposed over two years are compared to: (i) the uncracked reference samples (Figure 7a,b), and (ii) their corresponding samples tested after one-year exposure (Figure 7c,d).

The comparison of the cracked exposed samples against the uncracked reference samples (see Figure 7a for bending and Figure 7b for uniaxial tension) shows that, generally; samples with smaller crack opening (e.g., 0.15 mm) present a moderate increase in toughness after exposure, whereas samples with larger cracks (e.g., 0.3 mm CMOD) have a statistically significant drop in toughness after the exposure.

Results comparing samples tested after two-year exposure and one-year exposure (see Figure 7c for bending and Figure 7d for uniaxial tension) show generally a non-significant variation on the toughness with time for most of the samples. Furthermore, uniaxial tension samples showed a trend of higher toughness values with time for CMOD < 2 mm which decreased at the end of the tail (i.e., at CMOD = 4 mm), see Figure 7d; however, this was the case mostly for samples cracked at 0.15 mm.

These results show in general a negligible decrease of toughness of cracked SFRC exposed to wet-dry cycles compared to uncracked SFRC. In several cases, particularly for the samples cracked at 0.15 mm, the toughness of the exposes samples was significantly higher than the reference samples. However, there were two cases: (i) the samples exposed to 7 wt.% NaCl cracked at 0.3 mm showed a statistically significant decrease in work at larger deformations over time for both uniaxial tension and bending samples, attributed to the detrimental effect of fibre corrosion (discussed further in Section 3.3 and Section 3.4); and (ii) the exposed uniaxial tension reference sample cracked at 0.3 mm, that showed unexpectedly low values due to an unusual low fibre count in several specimens.

During the mechanical tests it was observed that, ca. 50% of the replicates from the exposed samples, showed branching and formation of new cracks close to the original crack (typically after first 0.5 mm CMOD), as shown in Figure 8. This phenomenon was not observed in uncracked reference samples and was more prominent in the samples cracked at 0.15 mm CMOD. This observation correlated well to the higher loads registered during the testing of cracked samples, and may explain the larger scatter in the results of the cracked samples relative to the uncracked references.

### 3.3. Exposure Conditions and Fibre Damage

The results from the fibre counting after one- and two-years are discussed below; and are based on the experimental results presented in Appendix B. The results are presented as the percentage of fibres classified by the deterioration levels described in Figure 3 (i.e., corrosion levels L1 to L3), relative to the depth inside the crack. The percentage of fibres rupturing instead of pulling-out is shown as “Fibre rupture” in blue. Results are shown for the bending samples in Figure 9 and uniaxial tension samples in Figure 10. An additional profile represents all fibres with severe corrosion: e.g., the combination of fibres with moderate and major corrosion (Levels 3–4). Finally, the profiles of uncorroded fibres (Level 1) and fibres with severe corrosion (Levels 3–4) from the samples tested after one year, are included as reference.

For this analysis, the outer 25 mm at the laterals of the crack and the fibres located at the compression zone of the cracked bending samples are omitted from the analysis in Figure 9, to consider just the corrosion extending from the crack mouth. So that the initial cross-section evaluated (150 × 125 mm) is reduced to an area of 100 × 90 mm for samples cracked at 0.15 mm CMOD and 100 × 100 mm for samples cracked at 0.30 mm CMOD.

Corroding fibres were mainly found at the outer crack area, the ratio of corroding fibres and degree of corrosion decrease gradually up to ca. 20–40 mm from the crack mouth, see Figure 9 and Figure 10. The extent and severity of fibre corrosion is generally larger for samples cracked at 0.3 mm compared to those cracked at 0.15 mm. Exposure to chlorides mainly entailed an increase in the severity of corrosion relative to the other exposed samples, i.e., larger share of fibres presenting moderate and major corrosion (Levels 3–4), see (Figure 9d–g and Figure 10d–g) and (Figure 9j,k and Figure 10j,k). In general, fibre rupture tended to increase at depths where most fibres present severe corrosion (Levels 3–4), i.e., the outer 10–20 mm of the crack depending on the exposure and crack width.

Comparison of results from samples tested after one- and two-years exposure generally showed a negligible progress of the extent of fibre corrosion over time for the samples cracked at 0.15 mm, for both bending (see Figure 9) and uniaxial tension samples (see Figure 10). Whereas the extent of fibre corrosion only increased significantly over time for some of the samples cracked at 0.3 mm exposed to 7 wt.% NaCl (w30s7c0), see Figure 9c and Figure 10c. Fibres showing signs of surface rust (i.e., minor corrosion L2) were also found over the sample cracked at 0.3 mm exposed to CO_2_ (w30s0c5), Figure 9i and Figure 10i.

### 3.4. Correlation of Fibre Corrosion and Fibre Rupture over Time

The correlation of corrosion damage of fibres bridging the crack to changes in the residual performance of the material over time was investigated by quantifying the impact of fibre corrosion on the quantity of fibres rupturing instead of pulling out.

The relation of fibre corrosion to the number of fibres rupturing is presented in Figure 11, as a boxplot showing the ratio of fibres ruptured at each corrosion level (L_1_–L_4_) and combinations of them. The ratio of ruptured fibres is compared at each corrosion level to a reference by a two-tailed Welch’s *t*-test with α = 10%; samples with a statistically significant higher ratio of ruptured fibres compared to the reference are marked in red in the figure. The figure describes the following analyses: (i) the percentage of fibres rupturing depending on their degree of corrosion in the samples tested after two-years exposure for bending (Figure 11a) and uniaxial tension samples (Figure 11b), using the ratio measured for non-corroded fibres (L_1_) as reference in the *t*-test; (ii) the total percentage of fibres rupturing depending on their degree of corrosion for the samples tested after one- and two-year exposure for bending (Figure 11c) and uniaxial tension samples (Figure 11d), indicating if there is a statistically significant increase in the ratio of ruptured fibres over time (i.e., using the data after one year of exposure as reference).

The first analysis, see Figure 11a,b, shows that for any degree of fibre corrosion there is a trend towards a significant increase in fibre rupture as corrosion is more severe, which corresponded well to the observations reported after one-year exposure (not presented herein). The contribution of fibres rupturing due to corrosion to the total count of fibres was significant for some cases, i.e., see Figure 11a,b.

The second analysis, see Figure 11c,d, shows that there is no significant increase in the ratios of fibres rupturing due to corrosion over time (i.e., L_2_, L_3_, and L_4_). However, there was a statistically significant increase in the contribution of corroding fibres to the total amount of rupturing fibres (i.e., combinations of L_1_ with L_2_, L_3_, and L_4_) over time for the uniaxial tension samples (see Figure 11d). This increase in the contribution over time was not observed for the bending samples, as shown in Figure 11c.

### 3.5. Corrosion Damage of Fibres and Mechanical Performance of Cracked SFRC

The relative impact of the exposure time on the mechanical performance of the cracked SFRC is discussed below by means of multiple linear regression (MLR). The MLR method was utilized to identify the main variables affecting the tensile toughness of the composite measured in bending and uniaxial tension tests and to quantify the contribution of these variables to the performance of the composite over time, based on the model described in Section 2.5. The variables used in this study are pre-selected based on a preliminary study utilizing the partial-least square (PLS) method [36] (not shown herein).

The MLR method is applied to the two datasets investigated in this study (i.e., bending and uniaxial tension data) in separate models. The model covers the quantification of the contribution of the following variables on the toughness of the composite up to a CMOD of 4 mm (y), being: the fibre content (x1), the ratio of fibres corroding (x2–x4), the ratio of fibres rupturing (x5), the crack width (x6) and the exposure time (x7). The linear predictors for the main variables and two-factor interactions are presented in Wilkinson notation for the bending tests in Equation (2) and for the uniaxial tension tests in Equation (3); where the non-statistically significant variables are presented between apostrophes ‘’ and non-significant interactions are omitted (i.e., considering α = 0.1).
y ~ ‘i’ + x1 + ‘x2′ + x3 + x4 + x5 + x6 + ‘x7′ + x2:x6 + x5:x6(2)
y ~ ‘i’ + x1 + ‘x2′ + ‘x3′ + x4 + ‘x5′ + x6 + ‘x7′ + x1:x4 + x4:x6 + x5:x6 + x6:x7(3)

The resulting coefficient estimates and p-values of the linear terms and coefficients of determination (R^2^ and adjusted R^2^) are presented in Table 3. Furthermore, the coefficient-estimates and normalized residuals (i.e., Standardized and Pearson residuals) of the resulting models are presented in Figure 12a,b for the bending tests and in Figure 12c,d for the uniaxial tension tests.

The coefficients of determination (R^2^) and the adjusted coefficients of determination are in the range 0.6 to 0.8, being slightly higher for the bending data. These values indicate an overall moderate fit of the data; however, the normalized residual plots, see Figure 12b,d, show a fair distribution with no signs of self-correlation, but with large normalized residual values; thus, indicating a large scatter in the experimental data.

The normalized coefficient estimates (z-scores) for the main predictors are presented in Figure 12a,c, which displays the relative impact of each variable on the toughness of the material. Positive estimates indicate an increase of the toughness when the variable increases. Non-significant estimate predictions, i.e., at α = 10%, are displayed in grey.

The coefficient estimates presented in Figure 12a,c show that the overall impact of the fibre content (x1) in the toughness of the cracked composite dominates over the relative impact of the other variables. A complementary assessment using the PLS method (not presented herein) showed identical results.

The relative impact of fibre corrosion (x2–x4) on the toughness of the composite estimated by the model shows a non-statistically significant positive impact of the fibres with minor corrosion (x2) and a negative impact of fibres presenting moderate and major corrosion (x3–x4), which was statistically significant for both groups in the bending samples (see Figure 12a) but only for the fibres presenting major corrosion (x4) for the uniaxial tension test (see Figure 12c). The ratio of ruptured fibres had a statistically significant positive relation to higher toughness values for the bending samples (see Figure 12a), and had a negligible impact on the uniaxial tension samples (see Figure 12b).

The crack width (x6) had a statistically significant negative impact on the toughness of the cracked composite for both datasets, see Figure 12a,c. Finally, the number of cycles (x7) did not have a statistically significant contribution in the residual toughness for none of the datasets see Figure 12a,c. Thus, suggesting that the expected negative contribution of the exposure time to the toughness of the cracked composite may be negligible at the time-scales investigated (i.e., comparing one and two years of exposure).

## 4. Discussion

This study investigated the impact of fibre corrosion on the mechanical performance of cracked SFRC exposed to wet–dry cycles of various corrosive environments over a period of two years. A summary of the results after two years of exposure to various environments is presented in Table 4, classified according to EN 206 as: limewater (XC0-1), 3.5 wt.% NaCl solution (XS3), 7.0 wt.% NaCl solution (XS3↑), fresh water and CO_2_ drying cycles (XC4) and 3.5 wt.% NaCl solution and CO_2_ drying cycles (XS3 + XC4). Toughness ratios calculated as the mean total work up to CMOD = 4.0 mm for each of the exposures relative to the reference samples tested after 56-days are given in Table 4.

Fibre corrosion was observed at the outer 10–40 mm of the crack (see Section 3.3), depending on the exposure and crack width, and did not vary significantly over time (i.e., after one-year exposure); for specific values refer to Table 4. Overall, the extent of fibre corrosion inside the crack increased mainly with larger initial crack width, while the presence of chlorides mainly increased the severity of fibre corrosion. The results suggest that fibre corrosion does not extend into the crack substantially with time, whereas the severity of fibre corrosion progressed gradually with time. Similar trends were found in previous research [11,16], yet former studies did not provide an accurate location and classification of the corroding and rupturing fibres.

It was observed that fibre corrosion had a clear impact on the proportion of fibres that ruptured instead of pull-out of the matrix (see Section 3.4). However, the overall increase of fibre rupture due to fibre corrosion was negligible in the largest cross-section investigated (e.g., 100 × 125 mm). Consequently, it is inferred that the impact of the exposure on the total number of fibres rupturing due to corrosion is strongly influenced by the size and shape of the specimen, since fibres corrode mostly at the outer 20–40 mm of the crack. For example, in the case of chloride exposure, corroding fibres were found at 75–100% of the of the cross-section of uniaxial tension test samples (80 × 80 mm exposed at all edges) but comprised only 20–30% of the cross-section of bending samples (150 × 125 mm exposed at three edges).

The results presented in Section 3.1 indicated negligible changes in the strength of the exposed uncracked concrete tested in compression over two years of exposure. These results substantiate that under these exposure conditions, neither corrosion of fibres in uncracked concrete nor deterioration of the concrete matrix due to exposure (e.g., chloride ingress, carbonation, and leaching) may have a significant impact on the strength of the bulk concrete matrix over time, in agreement with former studies [37,38,39].

Conversely, significant changes to the residual performance of the cracked material after exposure were reported in Section 3.2. There was a statistically significant increase in the toughness for smaller cracks (i.e., 0.15 mm) relative to the unexposed references. Whereas the samples cracked at 0.3 mm exposed to chlorides, showed a statistically significant drop in toughness over time, attributed to corrosion damage. Toughness ratios, calculated as the total work up to CMOD = 4.0 mm for each of the exposures relative to the uncracked references, are given in Table 4. Furthermore, additional cracking and branching during testing of the exposed specimens were attributed to an increase of the residual strength of the cracked SFRC (i.e., above the cracking strength). Insight on the topic suggests an increase of the fibre–matrix bond; as described based on former research in [4], and further investigated at both the single-fibre and composite levels on separate studies [40,41]. That increase in the fibre–matrix bond strength is expected to lead to localized stresses near the crack larger than the tensile strength of the adjacent uncracked matrix. Similar behaviour was reported for cracked round panels (i.e., crack width <0.1 mm) immersed in seawater and exposed to rainwater for 2 years [13].

Finally, the analysis presented in Section 3.5 indicates that the contribution of the variation of the total number of fibres crossing the crack generally dominates over the rest of the experimental variables; including the exposure time, which showed a negligible impact on the toughness of the cracked composite for the time-scale investigated. Overall, corrosion damage of the fibres had a negative, yet secondary, role when explaining variations in the residual performance of the cracked SFRC.

The discussion above indicates that there must be a differentiation in the cause of fibre rupture in cracked SFRC; fibres crossing the crack may rupture due to: (i) a critical reduction of the cross-section due to corrosion; or (ii) an apparent increase in the fibre–matrix bond strength over the exposure. Moreover, indicating that fibre rupture is not necessarily related to a decrease in toughness, i.e., described in this study as the total work; but may be the result of an unexpected increase of the fibre–matrix bond strength of the cracked composite during the exposure.

These arguments support the hypothesis that there are additional damage mechanisms besides corrosion damage of the fibres that explain the changes in the residual performance or the composite material [4]; for example, the alteration of the cement paste at the fibre–matrix interface, as described in [40,42] at the single-fibre level. A mechanism which was misattributed (in the authors’ opinion) in other studies to a higher roughness of the fibre due to corrosion [43,44].

This discussion presents evidence suggesting that variations observed in the mechanical performance of cracked SFRC under certain exposures may not be solely related to corrosion damage; so that the transport inside the crack and alteration of the matrix may have an important role to the mechanical performance of the cracked material.

### Engineering Implications

In general terms, the discussion in this paper indicates that the deterioration over time of the toughness of the cracked SFRC due to fibre corrosion is reduced, i.e., disregarding the contribution of the outermost fibres (e.g., at ca. the outer 10–40 mm of the crack), which could critically corrode, but may not compromise the long-term integrity of cross-sections larger than, e.g., 150 mm. These observations correspond well to some of the results reported in field exposure of cracked SFRC exposed to chlorides (i.e., EN-206 classes XS2 and XS3) [6,10,13,14,15,16] and carbonation (i.e., EN-206 classes XC4) [10,16] during periods of 1–5 year. However, results presented herein still disagree with conclusions from former studies, that predicted substantial decrease in residual performance over time in cracked SFRC due to fibre corrosion [11] or that measured a substantial decrease in toughness attributed to an excessive increase in the fibre–matrix bond strength, described as “embrittlement” [13,45].

Based on this investigation, fibre corrosion is only expected to have a negative impact on the residual performance of the material at the outer 10–30 mm of the crack surface, provided that there is a significant reduction of the fibre cross-section and a large share of the fibres critically corrode. Thus, emphasizing the impact of the specimen size in the deterioration observed. This may result in an overestimation of the exposure damage in very small specimens and would not be representative for the extrapolation to typical civil engineering applications: e.g., the cross-section thickness of a prefabricated segmental lining is in the range 200–400 mm [2,46], or is ca. 400–600 mm for a slab-on-grade [47]. However, the time-scale investigated in this study is substantially shorter compared to the typical aims for service life of such infrastructure, i.e., the design service life of a bored tunnel may be as long as 100–120 years [2,48,49], and further research is needed focusing on the assessment of existing structures built of SFRC.

Furthermore, the corelation observed between fibre rupture (e.g., below 20% of the fibres) and increase of toughness of the cracked material pointed-out to an unexpected increase of the fibre–matrix bond strength during the exposure. In this regard, an adequate selection of the fibre is critical to avoid general rupture of the fibres bridging the crack due to changes in the fibre–matrix bond strength over time and during exposure, as reported in [45,50]. For example, by selecting an appropriate steel strength and dimensions of the steel fibre, experimentally verified to the expected long-term strength class of the concrete matrix [4].

There is still limited data available from long-term studies that can be used to corroborate these observations, for example: inspection of SFRC infrastructure exposed to XS3 and XC4 environments during 20 years did not show substantial corrosion damage in steel fibres bridging small cracks, but did not provide any measure of the mechanical performance of the cracked composite [16]. Therefore, recommendations given in this paper may not be extrapolated to any design scenario or exposure time, since the discussion herein is still based on a limited number of experiments for short timescales and cannot be generalized to every type of fibre, concrete mix-design or exposure conditions.

## 5. Conclusions

This study comprised the exposure, mechanical testing, and inspection of uncracked and cracked SFRC specimens exposed to wet–dry cycles of chlorides and carbon dioxide for two years. The study focused on describing the extent of fibre corrosion over the exposure time and its impact on the mechanical performance of the SFRC. The following conclusions were drawn from the discussion herein:
The results presented in this study confirm that there is no substantial damage to uncracked SFRC exposed to wet–dry cycles of chloride and carbon dioxide, over the time-scale investigated. Fibres corroded primarily at the surface of the uncracked concrete and only produced aesthetical damage.Corrosion of steel fibres bridging cracks did not progress substantially inside the crack over a two-year exposure; and only entailed moderate reductions in the total toughness in small specimens cracked at 0.3 mm and exposed to large chloride concentrations. The moderate increase in the residual performance of the cracked SFRC at small deformations has been related to an increase of the fibre–matrix bond strength over time.The probability of fibre rupture increased with the amount of corrosion damage. But there was only a statistically significant increase in the contribution of fibre corrosion to the global count of ruptured fibres over time for small cross-sections. Investigations showed that that there was no clear detrimental relation between the number of ruptured fibres and the toughness of the material.Fibre corrosion had a subordinate, yet statistically significant impact on the changes in toughness over time, relative to the toughness variation observed due to the fibre distribution; while the impact of the exposure time was negligible, considering the time-scales investigated.

The results discussed in this paper do not indicate that fibre corrosion may have a critical impact on the bulk toughness of cracked SFRC over time for typical engineering applications. Whereas results indicate that there may be additional mechanisms responsible for some of the changes observed in the mechanical behaviour of the cracked SFRC after the exposure.

Further research focusing on describing these mechanisms and their effects (together with fibre corrosion) on the long-term performance of cracked SFRC is needed, preferably based on inspection of existing infrastructure.

## Figures and Tables

**Figure 1 materials-14-02642-f001:**
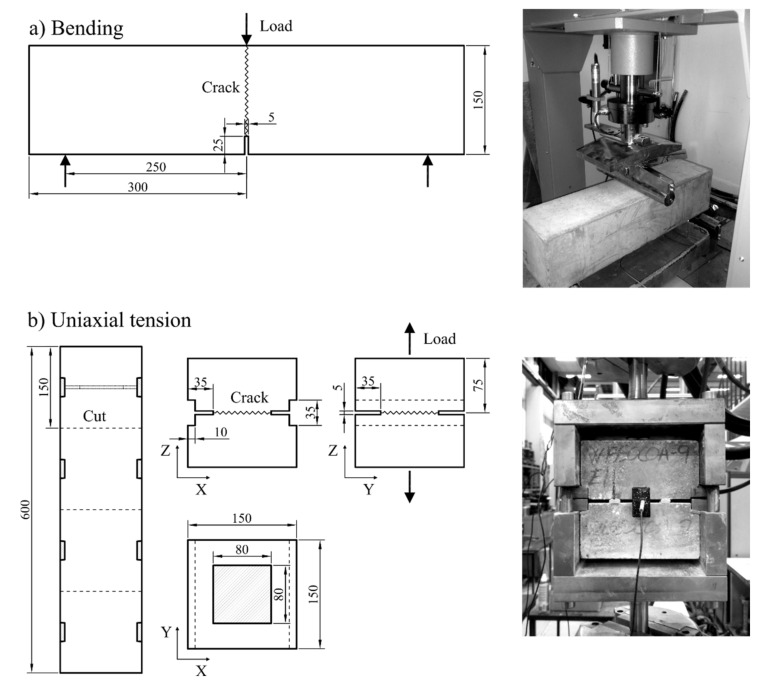
Dimension of specimens and test setups: (**a**) bending test, (**b**) uniaxial tension test. Dimensions are expressed in mm.

**Figure 2 materials-14-02642-f002:**
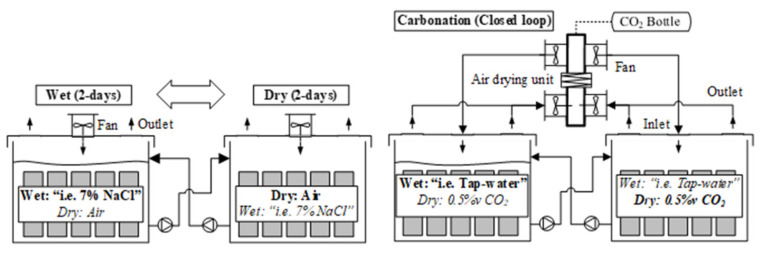
Exposure setup for wet–dry cycles.

**Figure 3 materials-14-02642-f003:**
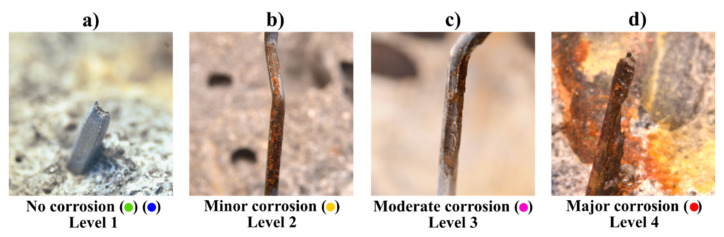
Fibre classification categories depending on fibre corrosion damage: (**a**) no corrosion (and fibre rupture), (**b**) minor corrosion, (**c**) moderate corrosion, (**d**) major corrosion (always fibre rupture). After [28].

**Figure 4 materials-14-02642-f004:**
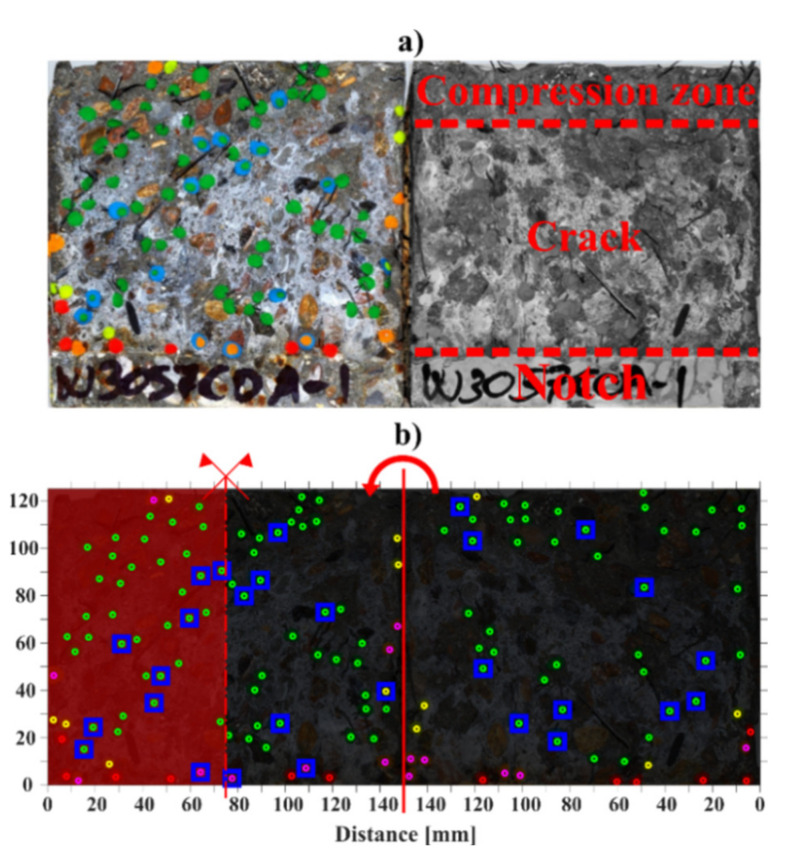
Visual inspection on the two faces of the crack of a bending specimen, fibre counting: (**a**) original cracked specimen with marked fibres and main features of the crack, (**b**) processed data on the two sides of the crack showing symmetry axes and final calculated area (highlighted in red). After [28].

**Figure 5 materials-14-02642-f005:**
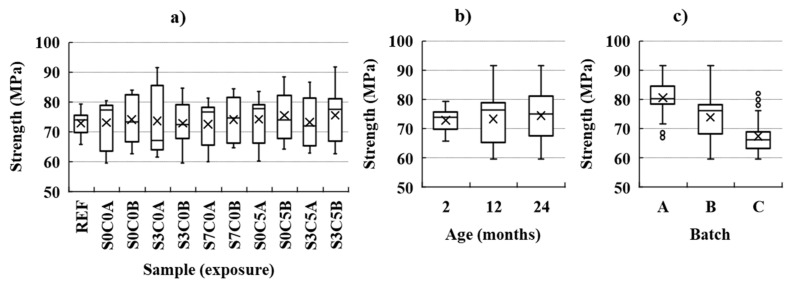
Compression tests results at reference and after one- and two-year exposure, grouped by: (**a**) exposure type, (**b**) age, and (**c**) batch. The mean (arithmetic) value is shown as “X”, the median value is shown as “─”, and outliers are shown as “o”. The “REF” group comprises the combined results from the unexposed reference samples tested after 56 days one- and two-years.

**Figure 6 materials-14-02642-f006:**
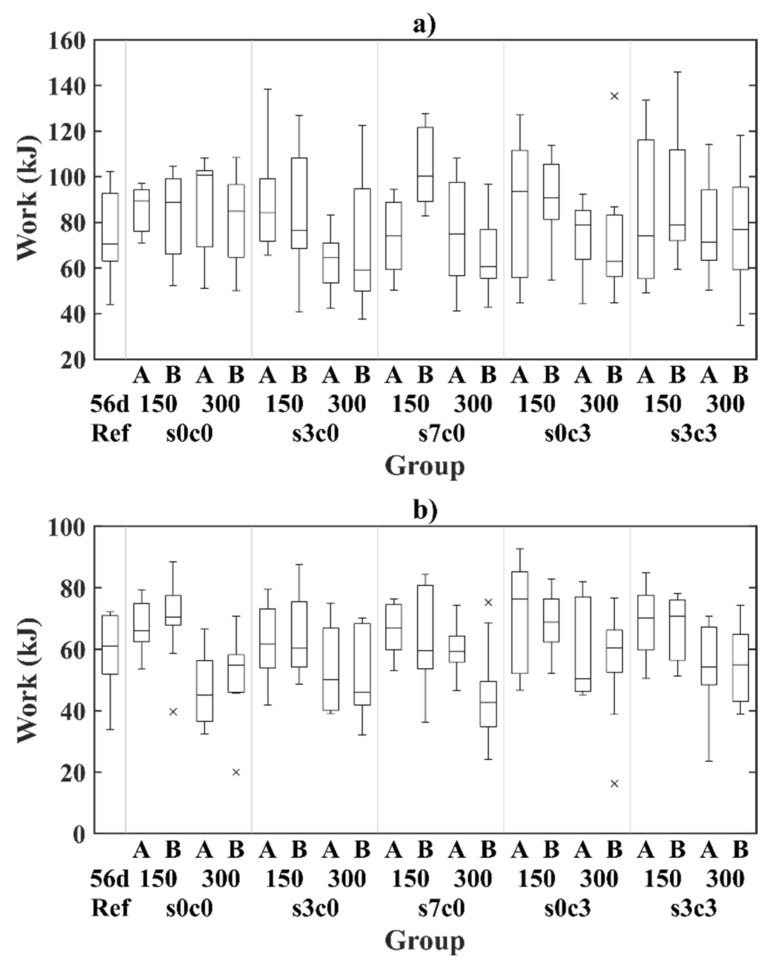
Boxplot, total work at 4 mm CMOD for the uncracked reference and exposed samples after one year (A) and two years (B) for: (**a**) bending test and (**b**) uniaxial tension test. Sample names correspond to code names described in Table 2.

**Figure 7 materials-14-02642-f007:**
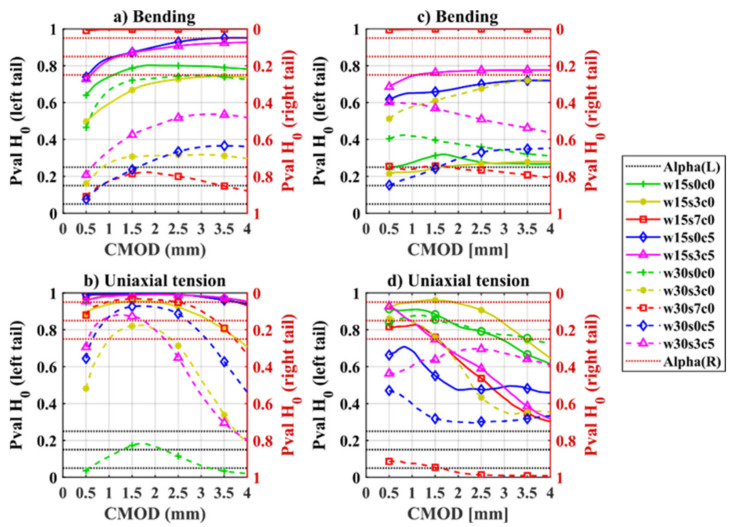
*t*-test, comparison of bending (top) and uniaxial tension (bottom) samples tested after two-years against: (**a**,**b**) uncracked reference samples, and (**c**,**d**) corresponding samples tested after one-year. Sample names correspond to code names described in Table 2.

**Figure 8 materials-14-02642-f008:**
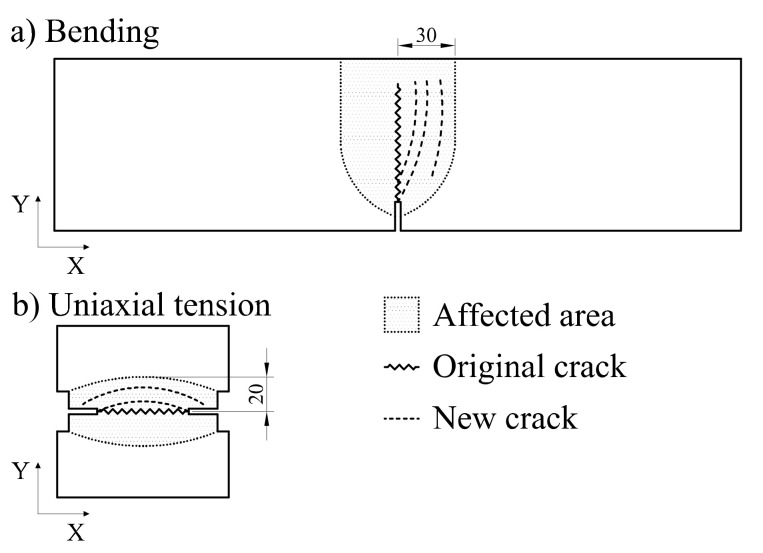
Typical crack path with additional branching for exposed specimens: (**a**) bending, (**b**) uniaxial tension. Dotted lines represent the additional cracks forming, describing the various crack patters observed. Dimensions are expressed in mm.

**Figure 9 materials-14-02642-f009:**
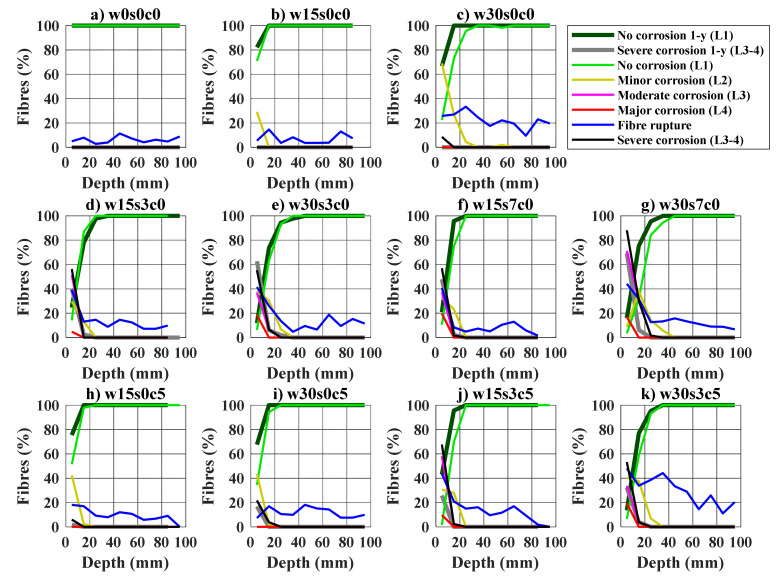
Fibre corrosion versus crack depth for bending samples: (**a**) uncracked reference, w0s0c0; (**b**,**c**) cracked reference, s0c0; (**d**,**e**) 3.5 wt.% NaCl exposure, s3c0; (**f**,**g**) 7.0 wt.% NaCl exposure, s7c0; (**h**,**i**) carbon dioxide and freshwater exposure, s0c5; (**j**,**k**) carbon dioxide and 3.5 wt.% NaCl exposure, s3c5. Sample names correspond to code names described in Table 2.

**Figure 10 materials-14-02642-f010:**
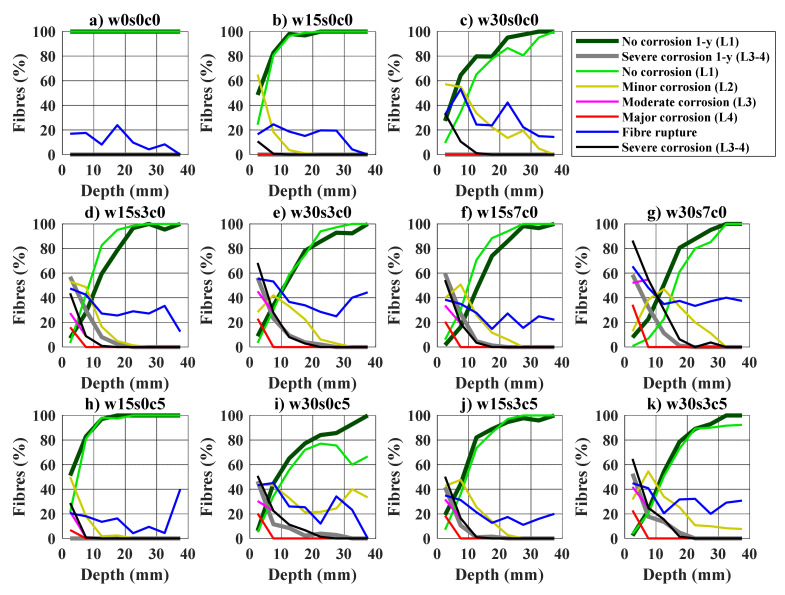
Fibre corrosion versus crack depth for uniaxial tension samples: (**a**) uncracked reference, w0s0c0; (**b**,**c**) cracked reference, s0c0; (**d**,**e**) 3.5 wt.% NaCl exposure, s3c0; (**f**,**g**) 7.0 wt.% NaCl exposure, s7c0; (**h,i**) carbon dioxide and freshwater exposure, s0c5; (**j**,**k**) carbon dioxide and 3.5 wt.% NaCl exposure, s3c5. Sample names correspond to code names described in Table 2.

**Figure 11 materials-14-02642-f011:**
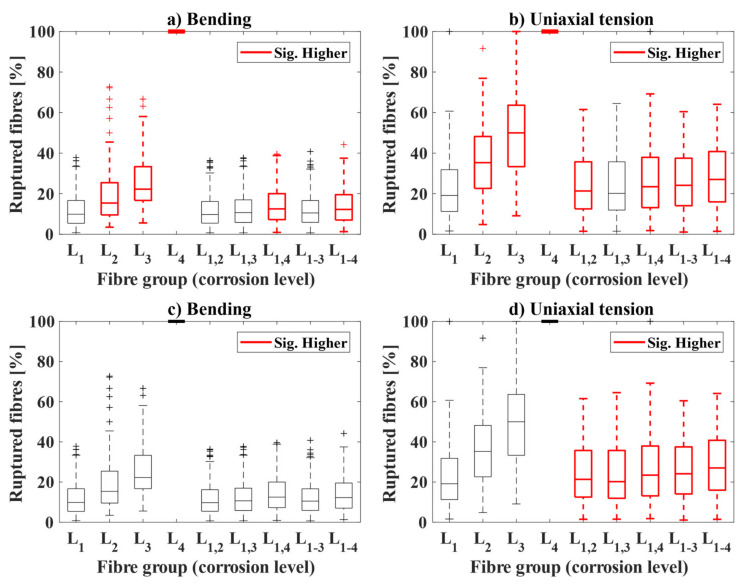
Impact of fibre corrosion on fibre rupture over time for: (**a**) bending samples after two years, (**b**) uniaxial tension samples after two years, (**c**) bending samples after one- and two-years (**d**) uniaxial tension samples after one and two years. Levels of fibre corrosion are represented by numbers L_1_–L_4_, based on the classification given in Section 2.4 (see Figure 3).

**Figure 12 materials-14-02642-f012:**
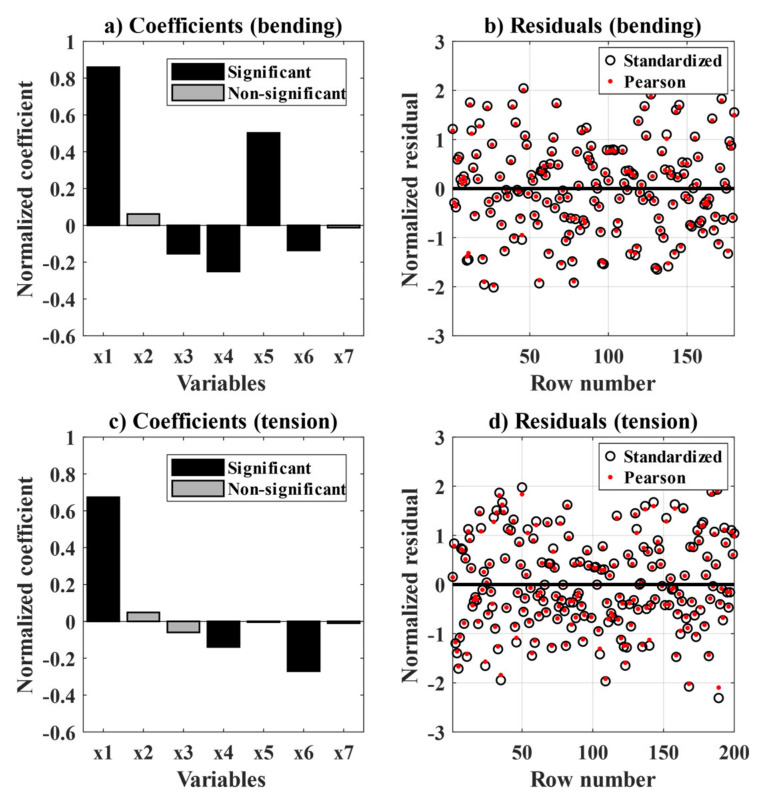
Multiple regression model for bending specimens and uniaxial tension specimens, showing: (**a**) normalized coefficients for bending specimens, (**b**) residuals for bending specimens, (**c**) normalized coefficients for uniaxial tension specimens, (**d**) residuals for uniaxial tension specimens.

**Table 1 materials-14-02642-t001:** Mix-design. After [28].

Component	Quantity (kg/m^3^)
Cement (CEM I 52.5N)	326.3
Fly Ash	100.0
Water	145.0
Sand 00/02	626.5
Sea gravel 04/08	175.1
Sea gravel 08/16	933.6
Steel fibres	40.0

**Table 2 materials-14-02642-t002:** Test samples and exposure conditions. After [28].

Code Name	Crack Width (w)	Wet Cycle (s)	Dry Cycle (c)	Age
(Sample)	(mm)			(A/B)
w0–s0–c0t0	uncracked (w0)	-	-	56 days
w0–s0–c0–A/B	uncracked (w0)	-	-	1/2 years (A/B)
w15–s0–c0–A/B	0.15 (w15)	Limewater (s0)	Air (c0)	1/2 years (A/B)
w30–s0–c0–A/B	0.30 (w30)			
w15–s3–c0–A/B	0.15 (w15)	3.5 wt.% NaCl (s3)	Air (c0)	1/2 years (A/B)
w30–s3–c0–A/B	0.30 (w30)			
w15–s7–c0–A/B	0.15 (w15)	7.0 wt.% NaCl (s7)	Air (c0)	1/2 years (A/B)
w30–s7–c0–A/B	0.30 (w30)			
w15–s0–c5–A/B	0.15 (w15)	Fresh water (s0)	0.5% vol. CO_2_ (c5)	1/2 years (A/B)
w30–s0–c5–A/B	0.30 (w30)			
w15–s3–c5–A/B	0.15 (w15)	3.5 wt.% NaCl (s3)	0.5% vol. CO_2_ (c5)	1/2 years (A/B)
w30–s3–c5–A/B	0.30 (w30)			

**Table 3 materials-14-02642-t003:** Multiple regression model results. Estimates and p-values for regressors and coefficients of determination.

Variable		Bending		Uniaxial Tension	
		Estimate	*p*-Value	Estimate	*p*-Value
i	Intercept	0.061	0.123	0.062	0.198
x1	Fibre count	0.859	0.000	0.674	0.000
x2	Corr. L2 (%)	0.062	0.157	0.049	0.367
x3	Corr. L3 (%)	−0.154	0.001	−0.060	0.348
x4	Corr. L4 (%)	−0.250	0.000	−0.139	0.027
x5	Rupture (%)	0.502	0.000	−0.004	0.952
x6	Crack width	−0.136	0.001	−0.270	0.000
x7	Exposure cycles	−0.012	0.784	−0.010	0.834
-	-	R^2^	0.788	R^2^	0.633
-	-	R^2^ adj.	0.776	R^2^ adj.	0.607

**Table 4 materials-14-02642-t004:** Summary of results.

Exposure (Sample)	Exposure Class (EN 206)	Crack Width (mm)	Corrosion Front Depth (mm)		Severe Corrosion Front Depth (mm)		Toughness Variation (-)	
			3PB	UTT	3PB	C	3PB	UTT
w15s0c0	XC1 (XC4)	0.15	20	20	-	10	1.10	1.19 *
w30s0c0		0.30	30	>40	10	10	1.01	0.87
w15s3c0	XS3	0.15	30	30	20	20	1.09	1.09
w30s3c0		0.30	40	40	30	20	0.90	0.87
w15s7c0	XS3↑	0.15	30	30	30	20	1.33 *	1.10
w30s7c0		0.30	40	40	30	30	0.81 *	0.78 *
w15s0c5	XC4	0.15	20	20	10	10	1.18 *	1.17 *
w30s0c5		0.30	30	>40	20	30	0.92	0.96
w15s3c5	XC4 + XS3	0.15	30	30	20	20	1.19 *	1.13 *
w30s3c5		0.30	40	>40	30	20	0.98	0.93

Abbreviations: (Exposure “sample”) codes of the experiment exposures according to Table 2, (Exposure class) corresponding exposure classes according to EN 206, (Crack width) crack width during the exposure expressed in mm, (Corrosion front depth) depth of corrosion inside the crack measured from the exposed edges for any level of corrosion expressed in mm, (Severe corrosion front depth) depth of severe corrosion “corrosion levels 3–4” inside the crack measured from the exposed edges expressed in mm, (Toughness variation) the ratio between the toughness of the sample tested after two-year exposure and the reference sample tested unexposed at 56 days, statistically-significant values (α = 0.1) are marked with “*”, (3PB) three-point bending specimens, (UTT) uniaxial tension test specimens, (C) compression specimens.

## Data Availability

The processed experimental data used herein may be found in Marcos-Meson, Victor (2021), “Mechanical performance and fibre counting of bending and tension SFRC specimens exposed to wet–dry cycles”, Mendeley Data, V1, doi: 10.17632/tjxp7njgt2.1.

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
