# Peer review of "Mechanical Performance of Steel Fibre Reinforced Concrete Exposed to Wet–Dry Cycles of Chlorides and Carbon Dioxide"

_materials, 2021, doi:10.3390/ma14102642_

Round 1
Reviewer 1 Report
Dear authors,
I am highly pleased and impressed by the extensive study you are submitting.
The topic is highly valued and mainly usable in research and practice.
I do not feel that there is a major problem in terms of test evaluation, because everything is well described. Perhaps only sometimes unnecessarily tedious - please take it as a recommendation - shorter articles bring better information for readers (scientists) in modern times.
The paragraphs are repeated (at least their meaning): for example, the last one in Chapter 4.1 and the last one in Chapter 5.
In conclusion, you write "Nonetheless, the direct extrapolation of these results to general design scenarios is still questionable, since the discussion currently is still based on a limited number of experiments under controlled conditions for a relatively short period of time compared to the expected service life of typical infrastructure. "
If I went through your published articles, I see that you have had to do many different tests and therefore the conclusions could be much more specific and may contain specific recommendations. You did analyze 420 specimens.
With all due respect, I would like to ask you to briefly explain the difference between the results presented and the results of your references (40, 42, 4, 35 and 36).
I'm aware of the notes on page 20, 10, etc., but I don't think that's enough.
The above mentioned samples (number of 420 pieces) are the same samples as in other articles?
The article is missing "Author Contributions:" "Conflicts of Interest:".
I feel that the numbering of the Appendix chapters should not be tied to the Conclusions.
Regards,
Author Response
We appreciate the positive review and very constructive comments. Please, find a reply to these (shown in bold) in the paragraphs below.
Apologies in advance for the very long reply, please let us know if some of these argumentation should be added to the revised manuscript. So far, we have tried to contain the length of the manuscript, since we acknowledge it is quite extensive.
I do not feel that there is a major problem in terms of test evaluation, because everything is well described. Perhaps only sometimes unnecessarily tedious - please take it as a recommendation - shorter articles bring better information for readers (scientists) in modern times.
We appreciate the positive comment and also understand that it is a lengthy paper. We agree, that in particular, the methodology section is very long and could be further reduced. However, due to the large extent of the investigations and various techniques used, it had been a challenge to reduce the length further without missing key information.
That had been also the main reason why we had moved the plots containing the experimental results to the appendixes, and focused the paper on the statistical analyses. Regarding the statistical analyses, we we also understand that these may seem sometimes a bit "tedious" but in our opinion these are one of the main strengths of this investigation, since it allowed us to have a more objective evaluation of the results.
In any case, we would be glad to reduce the level of detail in any sections and paragraphs that the reviewer may seem needed. Please, also note that Reviewer #4 has requested further details to be included in the methods section (section 2.3). We will do our best to keep these additional descriptions as short as possible
The paragraphs are repeated (at least their meaning): for example, the last one in Chapter 4.1 and the last one in Chapter 5. In conclusion, you write "Nonetheless, the direct extrapolation of these results to general design scenarios is still questionable, since the discussion currently is still based on a limited number of experiments under controlled conditions for a relatively short period of time compared to the expected service life of typical infrastructure. "
We agree that the last paragraph in the conclusion could be avoided, since it tends to describe again what had been discussed. Therefore, we have removed it and replaced with a more constructive concluding remark.
If I went through your published articles, I see that you have had to do many different tests and therefore the conclusions could be much more specific and may contain specific recommendations. You did analyze 420 specimens.
We agree that we based our conclusions in a reasonable number of experiments and appreciate the acknowledgement. Yet, please find our arguments for being cautious with our conclusions in this publication. We hope that you would agree.
In our opinion, this investigation (and the additional studies mentioned) have been very useful on helping us understand better the relations between fibre damage (due to corrosion) and the overall pull-out process (i.e. including the evolution of the fibre-matrix bond strength over time and exposure).
It is correct that in some of our other publications (also cited here) we have been able to describe additional mechanisms (such as the increase of the fibre-matrix bond strength) that may have been overlooked in the past. In our opinion, this could have led to potential missinterpretion of exposure results in some of the former studies (however it is quite challenging to prove that).
Furthermore we did focus on one mix-design (which we believe is quite relatable in the practice), but still cannot represent the wide variety of materials (e.g. binders, fibres etc.) that may be used nowadays.
We think that these results should still be used in the context of the current state-of-the-art, and also noting that there is published research that proposes other theories (e.g. pointing towards potentially larger deterioration over time). Therefore, we would rather avoid adding very very strong (and specific) arguments that could be misused our of the context of the investigation (e.g. still a limited exposure time under laboratory conditions).
All in all, and as stated in the conclusions, our opinion is that this investigation could serve as a very good basis to complement further field studies of existing structures under these type of exposures.
With all due respect, I would like to ask you to briefly explain the difference between the results presented and the results of your references (40, 42, 4, 35 and 36). I'm aware of the notes on page 20, 10, etc., but I don't think that's enough.
We have clarified the paragraphs briefly in the revised draft (see markup in yellow in the draft), but we would suggest avoid extending the discussion too much, since the paper is already quite long and those references already describe those effects in more depth.
Please, find a brief description of the scope and contents of these publications below:
Reference [4] corresponds to a literature review of former studies that investigated this topic (i.e. corrosion damage of SFRC exposed to chlorides and carbon dioxide). This reference was written and published before this investigation, and it does not contain a reference to the data presented in this paper. This first investigation served as basis for the elaboration of our initial research hypotheses (i.e. in the context of my PhD studies); there we were already aware of potential contradictions in former studies that were pointing towards overlooked "mechanisms" in the deterioration processes during exposure.
References [40] and [42] correspond to a separate study at the single-fibre level. These investigation were actually done on a separate set of specimens (single-fibre pull-out specimens). These specimens were made of a comparable concrete mix and used the same type of fibres (also additional types for further comparison) and were exposed to corresponding conditions. The scope of these studies was to further investigate the observations on the study discussed in this paper (mainly in connection to the increase of the fibre-matrix bond-strength over the exposure).
Reference [35], corresponds to a dataset. This is a publicly accessible link to the complete set of processed experimental data discussed in this paper. This dataset has been uploaded to allow other researches to further use these results in their studies. This reference is intended to be used in combination with this paper (and the data shown in the appendixes).
Reference [36] corresponds to my PhD thesis. The thesis is actually focusing on connecting these publications (and additional studies, e.g. ref. [32] and [41]). After this paper is published (we hope for the best), I will request permission to MDPI to append the published version of this paper in the thesis.
The above mentioned samples (number of 420 pieces) are the same samples as in other articles?
All these references were part of the same study, so the exposure conditions and materials were the same (to facilitate comparison). Also, some of the results from these studies come from companion specimens of the ones shown here and additional chemical studies in these after testing (e.g. chloride profiles, pH profiles, XRD, petrographic analysis etc.).
References [40][42] (single-fibre pull-out): No, but the mix-design and types of fibres are comparable - so that the studies can relate to each other.
Reference [32] (chemical-petrographic study): The study shows additional chemical and petrographic tests performed in these specimens (after the mechanical testing), and also additional companion specimens (under the same exposure).
Reference [35] (numerical model): This paper describes a model and uses data from companion specimens of the ones in this study for model validation.
The overall "goal" (in the whole PhD study) was to look at the SFRC corrosion problem from the different angles (e.g. chemical deterioration - mechanical performance) and scales (single-fibre level and composite level) and connect them trough a numerical modelling approach.
In this wider context, this paper is focusing on the mechanical performance at the composite scale.
The article is missing "Author Contributions:" "Conflicts of Interest:".
Noted. These forms had been filled-in during the submission, but were not included in the draft. These will be added to the final draft version.
I feel that the numbering of the Appendix chapters should not be tied to the Conclusions.
Noted. If we had understood well, the reviewer is proposing to move the appendix section to the next page.
If that is the case, we fully agree, and have added a page-break after the acknowledgments section.
Reviewer 2 Report
The presented study investigated the effects of fiber corrosion on the mechanical performance of cracked SFRC exposed to wet-dry cycles in various corrosive environments. The reviewer feels that the authors should be greatly appreciated to submit such a well-organized paper which involved a lot of work and the technical content is considered to be more than enough. As such, the reviewer has no hesitation in suggesting its acceptance for publication in Materials. Nevertheless, I would like to bring a few comments to the attention of the authors, which might increase the quality of the paper even further:
(1) The reviewer wonders how to control the crack width to be 0.15 and 0.3 mm accurately. What is the difference between CMODM and CMODN? Why is the value of CMODM (0.10±0.01 and 0.25±0.03 mm) less than the target one? What is the relation between the target width and the existing 5 mm wide notch? The reviewer probably understands the authors' meaning, but please be more specific. Could you provide some photos showing the processed specimens before exposure?
(2) The authors indicated that they showed the clip gages in Fig.1. But it is not found there. Thus it is better to provide actual photos for the bending and uniaxial tensile loading tests;
(3) Page 7, the last para.: there is no fiber rupture (blue) in Fig. 3d;
(4) The authors speculated why fibers crossing the crack ruptured. This is very interesting. But please explain a little bit why the fiber-matrix bond strength was increased over the exposure;
(5) Several minor typos:
P1, Para. 3: “…2 – 3 years of exposure. [Which] entailed moderate…”;
P21, Para. 2: “(e.g. at approx. [at] the outer 10 – 40 mm of the …”;
P21, Para. 3: “…the fibres critically corrode[. Thus, emphasizing] the impact …”. Note that there are similar punctuation errors in the manuscript that cause a sentence to be incorrectly divided into two parts.
In addition, the authors claimed that “This may result in an overestimation of the exposure damage in very small specimens …” (P21, Para. 3). According to the context, “small” should have been “large”? Anyway, the Discussion section is excellent and insightful.
Author Response
The authors appreciate very much the positive review and constructive comments. We have addressed the questions and comments below.
(1) The reviewer wonders how to control the crack width to be 0.15 and 0.3 mm accurately.
The specimens where cracked using a servo-controlled test-frame (using a crack-mouth opening displacement CMOD controlled test). Therefore, the force supplied by the hydraulic actuator (cross-head) was adjusted automatically (at a frequency resolution of approx. 1kHz in our case) using the clip-gage readings as the controlling parameter. The controller was calibrated during the pre-study trials to ensure a "smooth" CMOD displacement even after first cracking.
During the "pre-cracking" process, the frame was run at low CMOD rates (i.e. 50µm/min) until reaching the target CMOD value at the notch. At that point, the frame cross-head was set in "locked state" and the HDPE wedges were inserted into the existing notch. These inserts contributed maintaining the crack open (since the fibres tend to ). Then we left the specimen unloaded with the clips on, until the CMOD stabilized (after approx. 10-min the CMOD did not vary). This process had been fine-tuned during our pre-study trials (through a long trial-and-error process I'll admit), where we left some cracked specimens monitored with the clip gage after craking for 2-weeks (the CMOD did not change substantially after approx. 1-hour form testing).
We had also monitored the crack opening over the exposure (after 1- and 2-years) by measuring with a micrometer directly at the notch at reference locations on each specimen, we did not notice a measurable change over time (we had a lower measuring precision with this second "check" method: approx 20µm).
What is the difference between CMODM and CMODN?
These refer to the following:
CMODN: Refers to the CMOD value measured at the end of the notch (i.e. at the surface of the beam). This is the value that was "directly measured" during the test.
CMODM: Refers to the CMOD value calculated at the crack mouth. This value was calculated from the CMODN value. This is the actual value that corresponds to the mean maximum crack widht for the bending specimens, or just the mean crack width for the uniaxial tension specimens.
For the bending specimens, we reached to this value (derived from the value at the notch end by means of two studies (part of our pre-study preparations):
1- We used Digital Image Correlation to measure the actual crack-mouth value and the position of the crack tip (i.e. the cracked hinge) at the specimen trough the test. This was done on 10 specimens part of the pre-study.
2- We calibrated an in-hour numerical model (at DTU) based on the cracked-hinge model with the DIC data to calculate
For the uni-axial tension specimens, we did a similar process, but only involving the DIC measurements - to confirm that the average CMODN value measured by the two clip gages corresponded well to the actual crack width CMODM.
Why is the value of CMODM (0.10±0.01 and 0.25±0.03 mm) less than the target one?
The target CMODM opening was the target opening (at the crack mouth) during the pre-cracking test. However, after releasing the load once the pre-cracking was finished, we noticed that the crack would close slightly. We found out that there was no "practical" solution that would allow us to keep the crack open of that many specimens over 2-years.
Besides the HDPE inserts, we had tried blocking with an epoxy filler and metallic inserts, and also using a small loading-rig. We noticed that all of them resulted in the cracks closing a few µm, and we chose the HDPE inserts since were the most practical solution.
When using the HDPE inserts, we observed that the crack would close down by a few µm at approx. the first 10min after unloading the specimens. Therefore, we decided to just measure the actual "stable" crack after unloading, and use that value (in average).
What is the relation between the target width and the existing 5 mm wide notch?
The existing notch is just an "induced weakness" in the specimens, as as local reduction of the cross-section to ensure that the crack initiates at that specific location. The reason for that, is that the crack opening is only measured at that specific spot, and we aim at inducing only one crack.
The crack width (at the end of the notch) is what we are actually interested in our study. Note the following:
-In the bending specimens, the CMOD displacement measured at the end of the notch (i.e. at the specimen surface) (CMODN) is larger that the actual one measured at the crack mouth (CMODM). We had applied a correction factor for that in our tests, so that CMODM = k*CMODN (being k approx 0.72-0.76 - depending on the CMOD). We calibrated these based on the DIC experiments during our pre-study investigations.
- In the uniaxial tension specimens, the CMODM = mean(CMODN). Since we measure at two points at oposite sides of the centre.
The reviewer probably understands the authors' meaning, but please be more specific. Could you provide some photos showing the processed specimens before exposure?
We have included some pictures of the specimens in the test frames. Also, please find attached a quick figure with some annotated images showing the position of the inserts. Please, let us know if you would recommend to include such a figure in the next version of the draft.
We would be happy to include this explanation (or part of it) in the draft if the reviewer finds it essential. Our opinion is that it takes quite some explanation to show clearly the process and may shift the focus of the study and extend the methods section too much.
(2) The authors indicated that they showed the clip gages in Fig.1. But it is not found there. Thus it is better to provide actual photos for the bending and uniaxial tensile loading tests;
We appreciate the comment. This was actually a mistake in our effort to simplify and reduce the length of the paper. We have added the figures in Figure 1
(3) Page 7, the last para.: there is no fiber rupture (blue) in Fig. 3d;
We agree that the figure may lead to confusion. Actually, that fibre had ruptured further above, but we cropped the image at an unfortunate area. We have replaced the subfigure with a more comprehensive one.
(4) The authors speculated why fibers crossing the crack ruptured. This is very interesting. But please explain a little bit why the fiber-matrix bond strength was increased over the exposure;
That had actually been a large part of the parallel investigations that we have carried out during my PhD project (this publication is a part of it). In short, we found that there is a combination of two processes:
- Mainly, the re-precipitation of secondary phases (e.g. mainly calcite) at micro-cracks forming around the (partially pulled-out) fibres that bridge the main crack of the SFRC. Plainly, one could define it as "autogenous healing" of such micro-cracks at the fibre-matrix interface.
- Potentially, additional hydration reactions occuring at the bulk matrix. We did not find much evidence of this one, but is has been pointed out as an important aspect in former studies as well.
This aspect was dealt with in more depth in the publications below. We tried not not include too much of this discussion in this publication, to avoid repeating over the discussion on these and not extend this paper too much. Please, let us know if you would like to have further explanation in the draft.
V. Marcos-Meson, A. Solgaard, G. Fischer, C. Edvardsen, A. Michel, Pull-out behaviour of hooked-end steel fibres in cracked concrete exposed to wet-dry cycles of chlorides and carbon dioxide – Mechanical performance, Constr. Build. Mater. 240 (2020) 117764. https://doi.org/10.1016/j.conbuildmat.2019.117764.
V. Marcos-Meson, A. Solgaard, T.L. Skovhus, U.H. Jakobsen, C. Edvardsen, G. Fischer, A. Michel, Pull-out behaviour of steel fibres in cracked concrete under wet-dry cycles – deterioration phenomena, Mag. Concr. Res. (2020) 1–32. https://doi.org/10.1680/jmacr.19.00448.
V. Marcos-Meson, M. Geiker, G. Fischer, A. Solgaard, U.H.H. Jakobsen, T. Danner, C. Edvardsen, T.L.L. Skovhus, A. Michel, Durability of cracked SFRC exposed to wet-dry cycles of chlorides and carbon dioxide – Multiscale deterioration phenomena, Cem. Concr. Res. 135 (2020) 106120. https://doi.org/10.1016/j.cemconres.2020.106120.
(5) Several minor typos:
We appreciate the comments. We have addressed the comments in the revised manuscript, as explained below:
P1, Para. 3: “…2 – 3 years of exposure. [Which] entailed moderate…”;
Noted, the text has been corrected accordingly:
"...2 – 3 years of exposure. [The exposure] entailed..."
P21, Para. 2: “(e.g. at approx. [at] the outer 10 – 40 mm of the …”;
Noted, the text has been corrected accordingly:
"...(e.g. at approx. the outer..."
P21, Para. 3: “…the fibres critically corrode[. Thus, emphasizing] the impact …”. Note that there are similar punctuation errors in the manuscript that cause a sentence to be incorrectly divided into two parts.
We are not fully sure to understand the comment. Please, find a response with our interpretation of the comment:
If the reviewer refers to the comma after "Thus" (e.g. Thus,....). It is our understanding that in such a case (e.g. Thus at the beginning of a sentence) there should be a comma. Please, note that we wrote the draft using British English.
In addition, the authors claimed that “This may result in an overestimation of the exposure damage in very small specimens …” (P21, Para. 3). According to the context, “small” should have been “large”? Anyway, the Discussion section is excellent and insightful.
Agreed, we have rephrased the draft accordingly.

Reviewer 3 Report
The article covers the topic of the Mechanical Performance of Steel Fibre Reinforced Concrete Exposed to Wet-Dry Cycles of Chlorides and Carbon Dioxide.
In my opinion, article presents valuable content.
The subject and the supporting experiments are informative and present added value to the body of knowledge on the subject area. The manuscript has excellent cohesion. This is an interesting paper that deals with a timely topic and novel idea. The use of SFRC nowadays is prominent given the attractiveness of this material to various construction applications.
In my opinion, the authors have achieved to provide interesting research work. The research has a good quality and has the parameters to be attractive to other scholars and the construction industry.
Some minor remarks:
1. I suggest to add separated point 2 - Research significance - Please descibe here the main essence of the research. (Why presented
paper is so important? What is major innovation accent in presented studies?).
2. Please provide the cement content.
3. Please explain why you used fibers with a length of 60 mm in the amount of 40 kg / m3 of concrete mixture.
I suggest to add this description in the research significance part.
4. How was measured the ultimate tensile strength of fibres? If this is data is received from manufacturer, please add this information in the text.
5. How many specimens were performed? Why the standard deviations for the compressive strength of the specimens are so high, I suggest to explain this phenomena. What is more, please add some photos of the specimens.
6. I suggest that conclusions should be presented point by point.
Author Response
The authors appreciate the positive review and the constructive comments. Please, find a response to your questions and comments below:
1. I suggest to add separated point 2 - Research significance - Please describe here the main essence of the research. (Why presented paper is so important? What is major innovation accent in presented studies?).
As suggested, we have included a paragraph describing the significance of the study, right at the end of the introduction. The introduction already contained a short paragraph describing the scope of the paper, therefore we have combined both into a new subsection (subsection 1.1. Scope and research significance)
2. Please provide the cement content.
The cement type and content was already given in Table 1 (specified as kg/m3). We apologize if it was not sufficiently visible. Please, let us know if it shall be specified elsewhere.
3. Please explain why you used fibers with a length of 60 mm in the amount of 40 kg / m3 of concrete mixture. I suggest to add this description in the research significance part.
The main reason for selecting such a fibre length and dosage is that these are quite typical in prefabricated segmental lining production (one of the main applications for this material). Typically, characteristic strength classes (per fib model-code) of 4C (at 28-days) are specified for such applications.
The fibre length and tensile strength class correspond (also) to typical materials for this applicaton. The fibre type and strength class were selected in collaboration with the industrial partners of this investigation (part of my PhD) that cover both manufacturer/suppliers of steel fibres and engineering consultants.
We have included this clarification in the research significance paragraph (at the end of the introduction).
4. How was measured the ultimate tensile strength of fibres? If this is data is received from manufacturer, please add this information in the text.
The tensile strength of the fibres specified in the paper corresponds to the characteristic strength specified by the producer (confirmed with their records and by our tests in our lab). We have updated the information in the text.
Please, note that we are not supposed to state the fibre "brand" in the publication, since the study had been funded by a group of three steel fibre producers (Bekaert, Arcelor and Krampeharex). The fibre used was manufactured by Arcelor (as agreed internally in the project), but can be considered a fibre representative for the three producers.
5. How many specimens were performed? Why the standard deviations for the compressive strength of the specimens are so high, I suggest to explain this phenomena. What is more, please add some photos of the specimens.
The specimen production comprised approx. 600 specimens (in total). It was about 10m3 of concrete (excluding the trial castings). A large number of specimens were used in the pre-study, where we had optimized the techniques that we presented here. That pre-study took about 9-months (but it is unfortunately a lot of effort that does not really show in the publications :-))
The larger standard deviation of the compressive strength specimens obeyed to the production of these (in a precast factory). Such a wide scatter is expected in the actual industrial applications of SFRC. Other compression specimens produced in the lab castings (for our internal reference) showed much lower scatter, but were not fully representative to what one would expect to happen in the actual application (e.g. at a concrete production plant).
The main reason for considering the precast-factory specimens was to have a representative indication on how our observations of deterioration compared to "actual production" of concrete for engineering applications. The production was done in the same precast factory (with experience on SFRC precast element production) on consecutive days and with the same operators under our supervision.
Regarding the pictures, we have included a picture of each type of specimen in their corresponding test-frames in Figure 1, next to the diagrams.
6. I suggest that conclusions should be presented point by point.
We have modified the conclusions, so that now they are show by points. The last paragraph (with the closing statement and further research recommendations) has been left as a separate paragraph.
Reviewer 4 Report
The manuscript “Mechanical Performance of Steel Fibre Reinforced Concrete Exposed to Wet-Dry Cycles of Chlorides and Carbon Dioxide” studied the corrosion damage of carbon steel fibre reinforced concrete, which exposed to wet-dry cycles of chlorides and carbon dioxide for two years, and its effects on the mechanical performance of the composite over time. The authors have employed correctly the techniques. I have found the methodological approach correct and the results are properly interpreted. The presentation of the problem is clear, the results correctly presented and the conclusions well explained. This manuscript is well written and well structured, with a proper English grammar and syntax. From a scientific point of view the manuscript shows a high degree of soundness, supported by pertinent references. Finally, this article definitely further the research in this field. It can be published in the journal “Materials” in present form.
Author Response
The authors appreciate the positive review and recommendation for publication.
Round 2
Reviewer 1 Report
Dear authors,
Thank you very much for such an extensive and detailed response.
I fully understand your motivation and accept the explanation.
Due to other edits you've made based on other reviews, the article is fine.
I recommend thinking about the numbering of the appendix. Chapters in the appendix have the numbers 5.X, but should be bound to the letter A (B) or without a number in general.
Regards,
Author Response
We appreciate the constructive feedback and are glad to hear that the reviewer is satisfied with our explanations and changes.
Regarding the last comment received, we fully agree. We have changed the headings in the appendixes sections, so that they appear as (A."n") and (B."n").